# Estimating the Decoupling between Net Carbon Emissions and Construction Land and Its Driving Factors: Evidence from Shandong Province, China

**DOI:** 10.3390/ijerph19158910

**Published:** 2022-07-22

**Authors:** Mengcheng Li, Haimeng Liu, Shangkun Yu, Jianshi Wang, Yi Miao, Chengxin Wang

**Affiliations:** 1College of Geography and Environment, Shandong Normal University, Jinan 250358, China; mclisd@163.com (M.L.); 2020010092@stu.sdnu.edu.cn (S.Y.); 2021010091@stu.sdnu.edu.cn (J.W.); 2Collaborative Innovation Center of Human-Nature and Green Development, Universities of Shandong, Jinan 250358, China; 3Institute of Geographic Sciences and Natural Resources Research, Chinese Academy of Sciences, Beijing 100101, China; haimengliu@163.com; 4College of Resources and Environment, University of Chinese Academy of Sciences, Beijing 100049, China

**Keywords:** net carbon emissions, construction land, decoupling, driving factor, Shandong Province, China

## Abstract

Human activities and land transformation are important factors in the growth of carbon emissions. In recent years, construction land for urban use in China has expanded rapidly. At the same time, carbon emissions in China are among the highest in the world. However, little is known about the relationship between the two factors. This study seeks to estimate the carbon emissions and carbon sequestrations of various types of land based on the land cover data of 137 county-level administrative regions in Shandong Province, China, from 2000 to 2020.The study estimated the carbon emissions for energy consumption using energy consumption data and night-time light images, hence, net carbon emissions. The Tapio decoupling coefficient was used to analyze the decoupling between the net carbon emissions and construction land, and where the model for the decoupling effort was constructed to explore the driving factors of decoupling. The results showed that net carbon emissions in Shandong Province continued to increase, and the areas with high carbon emissions were concentrated primarily in specific districts of the province. The relationship between net carbon emissions and construction land evolved from an expansive negative decoupling type to a strong negative decoupling type. Spatially, most areas in the province featured an expansive negative decoupling, but the areas with a strong negative decoupling have gradually increased. The intensive rate of land use and efficiencies in technological innovation have restrained carbon emissions, and they have contributed to an ideal decoupling situation. Although the intensity of carbon emission and the size of the population have restrained carbon emissions, efforts towards decoupling have faded. The degree of land use has facilitated carbon emissions, and in recent years, efforts have been made to achieve an ideal decoupling. The method of estimation of net carbon emissions devised in this research can lend itself to studies on other regions, and the conclusions provide a reference for China, going forward, to balance urbanization and carbon emissions.

## 1. Introduction

Climate warming in the 21st century is one of the major challenges facing society and addressing climate change has become a global concern. It is unequivocal that human activities cause climate change, and carbon dioxide emissions are the main factor in this process [1]. According to the “Special Report on Climate Change and Land” released by the Intergovernmental Panel on Climate Change (IPCC) in 2019, agriculture, forestry, and other land types account for 23% of greenhouse gas emissions, and the carbon dioxide absorbed by natural land processes is almost equivalent to 1/3 of the carbon emissions released by fossil fuels and industry [2]. The use of land, its structure, and the intensity of use are closely related to socioeconomic development and this profoundly impacts the ecological environment and the well-being of humans in a region [3,4]. Therefore, rational regulation of land use to achieve carbon reduction goals becomes an important measure for sustainable development in countries around the world. With rapid urbanization worldwide, the expansion of construction land is inevitably accompanied by high carbon emissions due to energy consumption, and there is a contradiction between the two processes. Hence, exploring the relationship between carbon emissions and construction land is of great significance for green and low-carbon development and efficient land use; moreover, coordinating the relationship between the two processes is crucial for achieving sustainable human development [5].

The total carbon emissions for China in 2020 were 9.899 billion tons, accounting for 30.70% of global carbon emissions [6]. In September 2020, President Xi Jinping highlighted the following: “China is committed to ensuring carbon emissions peak by 2030 and achieving carbon neutrality by 2060” [7]. The “14th Five-Year Plan” of China proposes that the country’s ecological civil construction will focus on carbon reduction, thereby promoting a comprehensive green transformation of economic and social development [8]. At present, in China, the task of reducing carbon emissions involves both domestic and international pressures. With the acceleration of urbanization in China, the scale of urban construction land has expanded, and some areas have even experienced disorderly expansion [9,10]. Construction land has crowded out a large amount of ecological land covered with vegetation, thus weakening the carbon sink capacity of the earth’s surface. The contradictions between construction land and non-construction land have become increasingly prominent. Therefore, does the expansion or reduction of construction land have an impact on carbon emissions, and to what extent? What other factors affect carbon emissions? To what extent do these factors contribute to the decoupling between carbon emissions and construction land? China, as a big carbon emitter [11], has only 30 years from carbon peaking to carbon neutrality. Based on the above questions and objectives, it is particularly important and urgent to conduct in-depth research on the relationship between carbon emissions and construction land.

Land is an important vehicle for terrestrial ecosystems, and various land use types are interdependent and inter-constrained. The expansion or reduction of construction land will cause changes in other land use types, and carbon emissions and carbon sequestrations will also change accordingly [12,13]. Based on this, in terms of the acquisition of a carbon emissions dataset, this study has estimated the net carbon emissions from the perspective of carbon sinks and sources of land use, using the carbon emissions (sequestrations) of each land type and the carbon emissions for energy consumption associated with construction land. Thus, the net carbon emissions can be estimated more accurately and have a scientific basis.

In China, a country endowed with a vast territory and natural resources, economic development varies from one region to another, and carbon emissions also vary in space [14]. The country needs to take differentiated and refined carbon reduction measures according to local conditions. The administrative units at a county level constitute the third-level administrative region of China, and they are the basis for the delivery and distribution of power at a local level. Therefore, in terms of the scale of research, the smallest rank for the county-level administrative region was selected as the research unit. At the same time, the decoupling relationship between net carbon emissions and construction land was analyzed with the aid of the Tapio decoupling coefficient. Combined with the Kaya identity and the Log-Mean Divisia Index (LMDI), a decoupling effort model was constructed to explore further the factors influencing net carbon emissions and the driving effects of various factors on the decoupling of net carbon emissions and construction land. The findings of the study provide insights into the sustainable development of urbanization and carbon emissions in the county areas of the country. Furthermore, the findings will help guide local governments in formulating targeted carbon emission reduction measures and provide a scientific reference for China to achieve “carbon peaking and carbon neutrality” as soon as possible.

## 2. Literature Review

Carbon emissions from land use refer to the carbon emissions and carbon absorption due to the transformation or maintenance of the land type, which are usually measured by the carbon emissions (sequestrations) linked to land use [15,16]. Carbon emissions from energy consumption are those caused by the large amount of energy consumed by human activities on construction land [17,18], and which have been estimated by input-output and the IPCC’s emission factors [19,20,21,22]. Nevertheless, these methods fail to capture carbon emissions at a fine scale. Therefore, many studies have tried to estimate the carbon emissions from energy consumption using night-time light images. In 2000, Doll et al. were the first to confirm that there was a significant correlation between night-time light brightness and carbon emissions [23]. Then, Elvidge et al. investigated the relationship between night-time light brightness and the energy associated with carbon emissions in 200 countries around the world, thus providing an empirical reference for simulating carbon emissions with night-time light data [24]. Ambreen et al. found that the carbon dioxide concentration and night-time light brightness of cities in India were both high, while those of villages were both low, showing that night-time light images can be used as a basis for estimating carbon emissions [25]. In recent years, studies in China have begun to use energy consumption data and night-time light images to estimate carbon emissions. For example, Su et al. were the first to estimate the carbon emissions of construction land for cities at the prefecture level in China, based on the Defense Meteorological Satellite Program/Operational Linescan System (DMSP/OLS) night-time light images, this study made up for the incomplete statistics on national carbon emissions [26]. Xiao et al. used the DMSP/OLS global stable night-time light data to simulate the energy consumption at the provincial level in China, and provided a useful reference in monitoring and assessing provincial energy consumption [27]. Du et al. estimated the carbon emissions of 289 cities in China using DMSP/OLS, National Polar-orbiting Operational Environmental Satellite System Preparatory Project /Visible Infrared Imaging Radiometer Suite (NPP/VIIRS) night-time light images, and carbon emissions from energy consumption [28].

As a prelude to reducing carbon emissions, researchers have explored the relationship between carbon emissions and land use. An early study showed that frequent changes in land use were an important cause of increased carbon emissions [29]. Land use and land cover change (LUCC) are significantly associated with deforestation, the loss of farmland, and the expansion of built-up land [30]. The conversion of land from pasture or forest to arable land will lead to a large increase in carbon emissions [31]. With the development of urbanization, the further occupation of arable land for construction will also lead to a large amount of carbon emissions [32]. Studies have confirmed that there was a significant correlation between urban non-agricultural land and carbon emissions [33]. Controlling the expansion of construction land and reducing the area of arable land occupied by construction land could effectively control the increase in carbon emissions [34]. With acceleration of urbanization of land, China’s structure, intensity, and efficiency of land use have changed. Feng and Guo et al. studied LUCC and carbon emissions of cities and showed that the increase in construction land and the decrease in vegetation cover led to a significant decline in carbon storage [35,36]. Zhang et al. confirmed that there was a two-way causal relationship between the quality of land urbanization and carbon emissions, and the former had a negative impact on carbon emissions in various provinces [37]. Tang et al. believed that low-level industrial development and land use management promote the increase of carbon emissions at the extensive land use stage, however, high-quality industrial development and land use optimization lower carbon emissions at the intensive land use stage [38]. A study has also demonstrated that intensive land use could reduce carbon emissions more effectively than extensive land use [39]. For example, Stone, using data from 45 major U.S. cities, found that disorderly expanding cities generated more air pollutants than compact cities [40]. Makido et al. used the Landscape Pattern Index to measure the compactness of land use in 15 Japanese cities, revealing that the carbon emissions of compact cities were lower than cities that had undergone disorderly expansion [41].

The land is the carrier of terrestrial ecosystems. Extensive attention has been paid to the impact of land use/cover on carbon emissions. Research has demonstrated that there were significant differences in the carbon sink and carbon source capabilities between different land types, and the conversion from one type to another inevitably led to changes in carbon emissions [42]. Land use has become an important factor affecting the distribution of surface carbon dioxide concentrations [43]. The land is also the spatial carrier of human activities. Some researchers believe that carbon emissions are affected by factors other than changes in land use. In particular, the carbon emissions associated with construction land are the result of multiple factors [44]. Methods such as the Stochastic Impacts by Regression on Population, Affluence, and the STochastic Impacts by Regression on Population, Affluence, and Technology (STIRPAT) model, the multiple regression model, and grey correlation analysis have been used to analyze the factors influencing regional carbon emissions [45,46,47]. In 2004, Ang pointed out that the LMDI decomposition approach, which does not produce residual errors, is more suitable for factor analysis of time series than other methods [48]. Since then, this method has been used widely to explore the factors influencing carbon emissions [49,50]. For example, Hu et al. analyzed the carbon emission factors for 57 Belt and Road Initiative countries based on the LMDI model [51]. Alajmi et al. used the LMDI model in their analysis of the growth factors of carbon emission in Saudi Arabia [52]. The factors influencing carbon emissions, such as economic development, industrial structure, carbon emission intensity, intensive rate of land use, the rate of urbanization, energy structure, and the size of the population, have been selected for decomposition research [53,54,55].

To sum up, detailed theoretical and empirical studies have been conducted on carbon emissions and land use. It has been found that land use transformation is an important driver for the increase of carbon emissions; especially, the expansion of construction land has accelerated the carbon emissions associated with energy consumption. Some researchers believe that the proportion of land used for construction has a weaker impact on carbon emissions than the total energy consumed, the level of economic activity, and the rate of urbanization, etc. Some researchers believe that there was an inverted U-shaped relationship between urbanization of land and carbon emissions. These debates provide a solid foundation for this study. However, there are still some problems that need to be addressed at the outset. First, existing studies on carbon emissions should place more attention on energy-related carbon emissions and ignore the carbon sequestration effect of land cover. Second, existing studies on carbon emissions and land use are mostly limited to the national, provincial, or the municipal scale, and little attention is paid to the county-level emissions, which makes it difficult to accurately formulate carbon reduction measures. Third, among the studies on decoupling, the existing studies mostly analyzed the relationships between carbon emissions and economic growth [56,57,58,59], the decoupling relationship between China’s net carbon emissions and construction land remains unclear. Little is known about the impact of the expansion or reduction of construction land on carbon emissions and the driving factors of the decoupling between the two factors. The major contributions of this study are as follows. First, we factor in the carbon source and carbon sink capacities of different land types and conduct empirical research using net carbon emissions. Second, the research scale is expanded to account for the decoupling between net carbon emissions and construction land at the fine county scale. Finally, we combine the Tapio decoupling coefficient, the Kaya identity, and the LMDI model to build a systematic framework, which is used to analyze the factors influencing net carbon emissions and the driving effects of each factor on the decoupling between the two.

## 3. Materials and Methods

### 3.1. Study Area

The research used the county-level administrative units, the smallest administrative level in China, as the study area and selected specifically all county-level administrative regions in Shandong Province. The province, located on the eastern coast of China and in the lower reaches of the Yellow River, is an important transportation hub along the “Belt and Road” [60]. The province plays a leading role in ecological protection and high-quality development of the Yellow River Basin and has obvious advantages in its location for this research [61]. At present, Shandong Province is in the process of rapid urbanization including an expansion of urban construction land. Data from the sixth and seventh national censuses show that Shandong province has the second largest population in the country after Guangdong province for the period 2000–2020. In 2020, Shandong Province ranked third in the country in terms of total economic output, and the growth rate for the total value of imports and exports of goods was generally among the highest in the country, making the region one of the most promising and dynamic regions for economic development in the country. However, the economic structure of Shandong Province is dominated by industry, including heavy chemical industries. According to the “China Energy Statistical Yearbook” and the “China Emission Accounts and Datasets (CEADs)” [62], from 2000 to 2012, Shandong Province was ranked first in China in terms of carbon emissions from an energy consumption standpoint. From 2013 to 2020, it was second in the country below Shanxi Province, a major coal producer. As a province with a large economy, a large population, and large carbon emissions, the relationship between Shandong’s development and carbon emissions is very typical in China, and the task of realizing low-carbon development is clearly more challenging than that of other provinces.

County-level administrative units in China include municipal districts, county-level cities, counties, etc. As of 2020, Shandong province has 137 county-level administrative units, including 57 municipal districts, 27 county-level cities, and 53 counties. Considering that the administrative division of Shandong Province was adjusted during the study period to ensure the uniformity and accessibility of the study data, the study area was defined uniformly according to the administrative division of Shandong Province in 2020, as shown in Figure 1.

### 3.2. Data Sources

The remote sensing data on land use were obtained from the Landsat TM images of 2000, 2005, 2010, 2015, and 2020 provided by the Resource and Environment Science and Data Center, Chinese Academy of Sciences at a resolution of 30 m. The night-time light data were sourced from the National Geophysical Data Center of the National Oceanic and Atmospheric Administration. The energy consumption data and the standard coal coefficient were from the China Energy Statistical Yearbook (2001–2021). Other data came from the Shandong Statistical Yearbook (2001–20201), the Shandong Urbanization Development Report (2001–2021), and the China National Intellectual Property Administration (CNIPA).

### 3.3. Methodology

Carbon sinks refer to the storage and absorption of carbon, and forest land, grassland, water bodies, and unused land are carbon sinks. Carbon sources refer to the intensity of carbon emissions, and arable land and construction land are carbon sources [63]. Among them, the land type in this work refers mainly to the Status and Classification of Land Use (GB/T21010-2007), in combination with six first-level land categories, including arable land, forest land, grassland, water bodies, construction land, and unused land, based on analysis of the Landsat TM images. The methodology used is outlined in Figure 2.

#### 3.3.1. Estimation of Net Carbon Emissions

(1)Estimation of carbon emissions from land use

Carbon emissions from land use refer to carbon emissions and carbon sequestration caused by soil and vegetation of arable land, forest land, grassland, water bodies, and unused land. As previously described [64,65], the carbon emission (sequestration) coefficients for arable land, forest land, grassland, water bodies, and unused land were obtained according to the latitude, longitude, and geographical location of Shandong Province, which were 0.422 t/(hm^2^·a) for arable land, −0.644 t/(hm^2^·a) for forest land, −0.021 t/(hm^2^·a) for grassland, 0.248 t/(hm^2^·a) for water bodies, and −0.005 t/(hm^2^·a) for unused land using the following equation:(1)$$E_{x}=\sum L_{i}{\times \delta }_{i}$$
where *Ex* represents the carbon emissions from land use, *i* is the land type. *L_i_* refers to the area of each land type. *δ_i_* indicates the carbon emission (sequestration) coefficient of each land type.

(2)Estimation of carbon emissions from energy consumption

Carbon emissions from energy consumption are the carbon emissions caused by energy consumption from human activities undertaken on construction land [66]. Previous studies have revealed a significant correlation between the intensity of night-time light and energy consumption due to fossil fuel emissions, and economic activity; further, night-time light images could effectively reflect the intensity of human activities [67,68]. Therefore, as previously described [69,70], the carbon emissions from energy consumption at the county level were estimated by the carbon emissions from energy consumption and night-time light data in Shandong Province. The process was as follows, namely, the preprocessing and fitting correction of night-time light images. First, the DMSP/OLS night-time light images from 2000 to 2013 were reprojected, resampled, and cropped to obtain the light data of China’s administrative boundaries. Jixi City, with a relatively stable socioeconomic development, was selected as the invariant target area, and the satellite data of F162007 were used as a reference dataset to construct a regression model with other years. Intra-year fusion and inter-year continuous correction were performed. Second, the annual synthesis, reprojection, resampling, and cropping of the NPP/VIIRS night-time light images from 2012 to 2020 were carried out to make them match the spatial resolution of the DMSP/OLS data. Outliers in the invariant area were removed, and the intra-year fusion and inter-year continuous correction were performed. Finally, DMSP/OLS and NPP/VIIRS night-time light data at the county scale for Shandong Province in 2012 and 2013 were counted, respectively. The regression relationship between DMSP/OLS and NPP/VIIRS data at the county scale for 2012 and 2013 was constructed using the county as the calibration unit, with the DMSP/OLS data as the reference and the NPP/VIIRS data as the calibration object. Considering the accuracy at the county scale, a linear regression without an intercept was chosen. As can be seen from Figure 3, the regression equation was Y = 0.880X where Y is the DMSP/OLS night-time light data for 2012 and 2013 and X is the NPP/VIIRS night-time light data for 2012 and 2013. The *R*^2^ value was 0.886, which passes the 1% significance test, and the fit is good. The inter-year continuous correction was performed on the DMSP/OLS data from 2000 to 2013 and the NPP/VIIRS data from 2014 to 2020 was based on this regression equation. Based on the above calculations, the night-time light data at the county level scale in Shandong Province were obtained for 2000–2020.

Measurement of carbon emissions from energy consumption. Based on the carbon emission coefficient for energy consumption provided in the 2006 IPCC National Greenhouse Gas Inventory Guidelines [71], the carbon emissions from energy consumption in Shandong Province from 2000 to 2020 was calculated using the following equation:(2)$$E_{y}=\frac{44}{12}\times \sum E_{i}{\times \theta }_{i}{\times \mu }_{i}$$
where *E_y_* represents the carbon emissions from energy consumption. *E_i_* is the consumption due to various energy sources. *θ_i_* refers to the standard coal coefficient of various energy sources. *μ_i_* indicates the carbon emission coefficient for various energy sources. The standard coal coefficient and carbon emission coefficient for various energy sources are shown in Table 1. The unit of heat converted into standard coal is kg standard coal/million J. The unit of electricity conversion is kg/kW h.

Estimation of carbon emissions and accuracy test. The total night-time light and carbon emissions for energy consumption in Shandong Province were fitted to build linear, logarithmic, quadratic polynomial, power exponential, and exponential fitting equations. To ensure the reliability of the fitted carbon emission values, the Mean Relative Error (MRE) of the carbon emissions in Shandong Province fitted by each equation and the carbon emissions from energy consumption calculated from the statistical data were calculated, that is, the MRE at the provincial scale. The night-time light values for the various regions were substituted into the regression equation. Considering the accuracy at the county scale, the regression equation without intercept was used for calculation. The weighted summation of the fitted carbon emissions in each region was used to obtain the fitted value of carbon emissions in Shandong Province, which was then tested for accuracy with reference to the carbon emissions from energy consumption, that is, the MRE at the county level scale, as shown in Table 2.

Table 2 indicates that the regression coefficients of the five fitting equations were all significant at the 1% level, where *R^2^* was greater than 0.65, indicating a good degree of fit. The accuracy test at the county level indicated that the MRE for carbon emissions fitted by the quadratic polynomial and the power exponential equations and that calculated by the statistical data were too large, while the carbon emissions for some areas fitted by logarithmic and exponential equations already exceeded the carbon emissions calculated by the province’s statistical data, which clearly was not possible and thus was excluded. The MRE for the carbon emissions fitted by the linear regression equation and the carbon emissions calculated by the statistical data were smaller, showing better stability. Thus, it is concluded that the nonlinear regression equation was not suitable for estimating carbon emissions at the county level. Therefore, a linear regression equation was used to estimate carbon emissions from energy consumption.

#### 3.3.2. Tapio Decoupling Coefficient

In 2005, when exploring the relationship between greenhouse gas emissions and economic growth in Europe, Tapio proposed the use of an alternative decoupling coefficient [72]. Compared with the Organization for Economic Co-operation and Development (OECD) decoupling coefficient, the Tapio decoupling coefficient overcomes the problem of base period selection and measures the decoupling relationship more accurately. In recent years, it has been widely used to explore the relationship between socioeconomic development and the environment [73,74]. Therefore, the Tapio decoupling coefficient was used to analyze the relationship between net carbon emissions and construction land using the following equation:(3)$$T=\frac{CE}{LC}=\frac{\left({CE}_{n}-{CE}_{O}\right){/CE}_{O}}{\left({LC}_{n}{-LC}_{O}\right){/LC}_{O}}$$
where *T* represents the decoupling coefficient. Δ*LC* refers to the rate of change in construction land. Δ*CE* indicates the rate of change in net carbon emissions. *CE_n_* and *LC_n_* represent the net carbon emissions and the area of construction land at the end of the study. *CE_O_* and *LC_O_* stand for the net carbon emissions and area of construction land for the base period of the study.

Tapio has classified decoupling relationships into eight categories according to the decoupling coefficient, as shown in Figure 4. Strong decoupling means that *CE* is declining while the *LC* is growing. Recessive decoupling means that both the *LC* and *CE* are declining, and the rate of decline for *LC* is slower than that for *CE*. Weak decoupling means that both the *LC* and *CE* are growing, and the rate of growth for *LC* is faster than that for *CE*. Strong negative decoupling means that *CE* is growing while *LC* is declining. Expansive negative decoupling means that both the *LC* and *CE* are growing, and the growth rate for *LC* is slower than that for *CE*. Weak negative decoupling means that both the *CE* and *LC* are declining, and the rate of decline of *LC* is faster than that for *CE*. Expansive coupling means that *LC* and *CE* are growing at similar rates. Recessive coupling means that *LC* and *CE* are falling at a similar rate.

#### 3.3.3. Decoupling Effort Model

In 1989, Yoichi Kaya proposed the Kaya identity at the IPCC International Symposium, which established an identity relationship between carbon dioxide produced by human activities and factors such as the economy and population [75]. Human activities are the main source of carbon emissions. Based on the Kaya identity and the LMDI model [76], a correlation between net carbon emissions and eight factors, including the intensity of carbon emissions, economic scale, the rate of intensive land use, industrial structure, the efficiency of technological innovation, the intensity of technological innovation, the size of the population, and the degree of land, was used to explore the impact of each factor on net carbon emissions. Carbon emission intensity ($\beta_{1}$) refers to the carbon emissions released per unit of economic output and is an important indicator of the relationship between economic development and the environment, characterizing the impact of each region’s level of economic development on net carbon emissions. The economic scale ($\beta_{2}$) was a measure of the economic output generated per unit of built-up land and measures the impact of economic concentration and development on net carbon emissions. The rate of intensive land use ($\beta_{3}$) is expressed as the area of land used for construction per unit of non-agricultural output and measures the impact of the economy and the intensive use of the resources of construction land on net carbon emissions. Industrial structure ($\beta_{4}$) is the basis for the orderly operation of regional production and life. Secondary and tertiary industries are important sources of carbon emissions. The measure of industrial structure is based on the impact of industrial structure on net carbon emissions, and it was calculated using the share of the value of non-agricultural industrial output. As data on science and technology innovation at the county level scale were difficult to obtain, this study relied on the number of patents granted to indirectly characterize the financial investment in science and technology innovation, the investment in research personnel, and other related inputs [77]. Among them, the technological innovation efficiency ($\beta_{5}$) gives a measure of the impact of the investment benefit of scientific and technological innovation on net carbon emissions using the economic output representation of the unit of patent authorization. The technological innovation intensity ($\beta_{6}$) was characterized by the per capita patent authorization amount to characterize the impact of the level of scientific and technological innovation on net carbon emissions. The size of the population ($\beta_{7}$) was used to characterize the impact of population density on net carbon emissions by measuring the number of people living on a unit of construction land and the regional differences in the population distribution. The intensity of land use ($\beta_{L}$) was a measure of the direct impact on net carbon emissions of the expansion or reduction of construction land during rapid urbanization, using the built-up area as a proxy as follows. The equation for the Kaya identity is:(4)$$C=\underset{i}{\sum }{(\beta }_{1}{\times \beta }_{2}{\times \beta }_{3}{\times \beta }_{4}{\times \beta }_{5}{\times \beta }_{6}{\times \beta }_{7}{\times \beta }_{L})=\underset{i}{\sum }\frac{C_{i}}{{GDP}_{i}}\times \frac{{GDP}_{i}}{L_{i}}\times \frac{L_{i}}{{GDP}_{i}^{*}}\times \frac{{GDP}_{i}^{*}}{{GDP}_{i}}\times \frac{{GDP}_{i}}{I_{i}}\times \frac{I_{i}}{P_{i}}\times \frac{P_{i}}{L_{i}}{\times L}_{i}$$
where *C* is the net carbon emissions for Shandong Province. *C_i_* refers to the net carbon emissions for the region *i*. *GDP_i_* means the gross domestic product of the region *i*. *L_i_* is the area of construction land for the region *i*. *GDP_i_** is the value of the non-agricultural output for region *i*. *I_i_* is the number of patents granted for the region *i*. *P_i_* is the number of people resident in the region *i*.

To further explore the driving effect of various factors on the decoupling relationship, the decoupling effort model was constructed in combination with the Tapio decoupling coefficient and the LMDI model [78], as indicated in Figure 5.

In the above flowchart, Δ*β*_1_, Δ*β*_2_, Δ*β*_3_, Δ*β*_4_, Δ*β*_5_, Δ*β*_6_, Δ*β*_7_, and Δ*β_L_* refer to changes in the carbon emissions due to changes in the intensity of carbon emissions, the economic scale, the rate of intensive land use, industrial structure, the efficiency of technological innovation, the intensity of technological innovation, the size of the population, and the degree of land use, respectively. Δ*C* stands for changes in carbon emissions due to the total effects. *T* and *o* represent the corresponding variables at the end of the study and the base period, respectively. A is the decoupling effort index. Δ*α*_1_, Δ*α*_2_, Δ*α*_3_, Δ*α*_4_, Δ*α*_5_, Δ*α*_6_, Δ*α*_7_ and Δ*α_L_* represent the decoupling effort index for the intensity of carbon emissions, the economic scale, the rate of intensive land use, industrial structure, the efficiency of technological innovation, the intensity of technological innovation, the size of the population, and degree of land use, respectively. Δα refers to the total decoupling effort index. If *α* ≤ 0, it means no decoupling efforts; in other words, the driving factor does not promote decoupling, but it does increase carbon emissions. If *α* ≥ 1, this means strong decoupling efforts are in play, and the driving factor promotes ideal decoupling. If 0 < *α* < 1, this indicates weak decoupling efforts, and the driving factors promote decoupling, but facilitation is weaker than that of the changes in construction land.

## 4. Results

### 4.1. Spatiotemporal Evolution of Land Types

The spatial evolution pattern for the different land types in Shandong Province from 2000 to 2020 was obtained based on analysis of the Landsat TM images. As shown in Figure 6, the area of construction land was second only to arable land, and its proportion increased by 4.772% from 2000 to 2020. Spatially, this expansion happened mainly along the Qingdao-Jinan Railway, the Yellow River Delta, and the coastal areas of the Shandong Peninsula. The most remarkable expansion was observed from 2005 to 2010. From 2015 to 2020, the area of construction land was reduced in the Yellow River Delta and the coastal areas of Binzhou, indicating that the wetland restoration project concerning aquaculture in these areas was an immediate success. Arable land is the main land type in Shandong Province. Its share decreased by 1.984% from 2000 to 2020, and arable land became fragmented spatially. Forest land and grassland were distributed mostly in the mountainous and hilly areas of central, southern, and eastern Shandong, from 2005 to 2015, and both forest land and grassland were degraded, especially the grassland in the Yellow River Delta. In recent years, affected by the policy of “returning farmland to forest land and grassland”, forest land and grassland expanded slightly from 2015 to 2020. From 2000 to 2020, the proportion of water bodies increased by 1.914%, mainly in Laizhou Bay, the Yellow River Delta, and the coastal areas of Binzhou. The unused land was distributed mainly in the Yellow River Delta, whose proportion decreased by 0.807% from 2000 to 2020.

### 4.2. Spatiotemporal Evolution of Net Carbon Emissions

The net carbon emissions for Shandong Province were obtained by estimating the carbon emissions from land use and energy consumption. As shown in Table 3, the net carbon emissions increased gradually from 2000 to 2020, and where the carbon sinks of forest land accounted for more than 70% of the total, indicating that the carbon sink capacity was strong; the grassland, water bodies, and unused land had fewer carbon sinks; the carbon emissions released by construction land accounted for more than 95% of the total carbon emissions, and represented the main source of carbon emissions, whereas the carbon emissions of arable land declined but with some fluctuation. From 2000 to 2020, the proportion of the total carbon sources that could be offset by carbon sinks fluctuated between 0.1% and 0.5%, and the total carbon sources far exceeded the carbon sinks.

The net carbon emissions for various regions in Shandong Province were classified manually into five levels: weak emission, low emission, medium emission, high emission, and strong emission. As shown in Figure 7, the increase in the levels of carbon emission from 2000 to 2020 first appeared in the municipal districts, resulting in the formation of carbon emission hotspots centering on municipal districts. The spatial distribution tended to cluster from points into surfaces over time, and the areas with low emission and medium emission gradually increased, thus forming two high emission centers in Jinan and Qingdao.

### 4.3. Decoupling between Net Carbon Emissions and Construction Land

#### 4.3.1. Temporal Evolution of the Decoupling Relationship

The decoupling between net carbon emissions and construction land in Shandong Province was calculated using Equation (3). Table 4 demonstrated that the decoupling between net carbon emissions and construction land in 2000–2005, 2005–2010, and 2010–2015 was expansive negative decoupling. In other words, both the area of construction land and net carbon emissions increased, but the net carbon emissions increased faster than the area of construction land. From 2015 to 2020, the decoupling evolved into a strong negative decoupling, indicating that the construction land decreased in Shandong Province in recent years, but carbon emissions were still increasing. In general, the decoupling between net carbon emissions and construction land from 2000 to 2020 was expansive negative decoupling. This showed that the relationship between net carbon emissions and construction land was unstable, and this was not conducive to the realization of carbon peaking and carbon neutrality goals. However, the reduction in construction land has helped to reduce carbon emissions and the decoupling relationship has begun to change.

#### 4.3.2. Spatial Evolution of the Decoupling Relationship

To analyze the spatial evolution of the decoupling between the net carbon emissions and construction land in Shandong Province, the decoupling in the five time periods was characterized spatially. As shown in Figure 8, the decoupling between the net carbon emissions and construction land in most areas of the province from 2000 to 2005 was expansive negative decoupling. From 2005 to 2010, the areas with expansive negative decoupling decreased, while the areas with strong decoupling and weak decoupling increased significantly, and most of them were concentrated in municipal districts. As shown in Figure 6, at this stage, the construction land in Shandong Province began to expand substantially from the center of municipal districts to the periphery. However, the speed of infrastructure construction and economic development brought about by the rapidly expanding construction land did not match it, and there was a lag in carbon emissions from energy consumption, thus enabling these regions to show an ideal decoupling. From 2010 to 2015, the areas with strong decoupling and weak decoupling decreased, and the growth rate of net carbon emissions in most areas of the province was still faster than that of construction land. From 2015 to 2020, the areas with expansive negative decoupling still dominated; however, the areas with strong negative decoupling expanded significantly, and these were concentrated mainly in the Yellow River Delta. This indicated that the construction land in these areas had begun to shrink, but the net carbon emissions were still increasing in recent years. In general, there was an expansive negative decoupling between net carbon emissions and construction land from 2000 to 2020 in most parts of the province. Net carbon emissions increased with the expansion of construction land, and the growth rate of net carbon emissions was faster than that of construction land. In addition, Hekou District, Zhanhua District, Wudi County, Lingcheng District, Ningjin County, Linyi County, Dong’e County, Gaoqing County, Shouguang City, Changyi City, and Taierzhuang District exhibited a strong negative decoupling. Licang District in Qingdao City had a weak decoupling, meaning that the growth rate of net carbon emissions was slower than that of construction land. Shinan District and Shibei District demonstrated a strong decoupling, with an expansion of construction land and a reduction of net carbon emissions. It can be seen that net carbon emissions in these areas have not increased in line with the expansion of construction land. As the first low-carbon pilot city in Shandong Province, Qingdao has witnessed, in recent years, a significant decline in carbon emissions.

### 4.4. Driving Factors of Decoupling between Net Carbon Emissions and Construction Land

#### 4.4.1. Factors Influencing Net Carbon Emissions

The LMDI model was used to measure the impact of various factors on net carbon emissions, and this was expressed by the degree of the contribution. As shown in Figure 9, the contribution of the economic scale was the largest throughout the five periods of the study, which promoted an increase in carbon emissions. This indicated that Shandong Province was still in a mode of economic growth with high growth and high energy consumption. The contribution of the intensity of technological innovation was also high but tended to decline, suggesting that research and development of low-carbon technologies were insufficient. This indirectly promoted an increase of carbon emissions, but the promoting effect gradually weakened. The inhibitory effect on carbon emissions by improvements in the efficiency in technological innovation tended to increase, which showed that the improvement in research efficiency was conducive to carbon reduction. The rate of intensive land use always inhibited the growth of carbon emissions, but its role gradually diminished. The intensity of carbon emissions evolved from an inhibitory effect to a promoting effect from 2015 to 2020, showing that the economic development of Shandong Province increased carbon emissions in recent years. The industrial structure developed from a promoting effect to an inhibitory effect from 2015 to 2020, indicating that the adjustment and upgrading of Shandong’s industrial structure inhibited carbon emissions, but the effect was weak. The inhibitory effect of the size of the population on carbon emissions gradually weakened and evolved into a promoting effect from 2015 to 2020. The continuous agglomeration of the population was an important reason for the increase in carbon emissions. The degree of land use changed from a promoting effect to an inhibitory effect from 2015 to 2020, and the reduction of construction land had an effect on carbon reduction.

From the perspective of having cumulative effects, the carbon emissions promoted by the economic scale were roughly offset by the carbon emissions inhibited by the efficiency of technological innovation and the rate of intensive land use, such that the three factors reached a dynamic balance. As such, it is necessary to further reduce carbon emissions by continuously improving the efficiencies of technological innovation and the rate of intensive land use. The carbon emissions promoted by the intensity of technological innovation could be offset by the carbon emissions inhibited by efficiencies in technological innovation, and the carbon emissions were further reduced on the basis of this offset. This indicated that the development of scientific and technological innovation had an increased inhibitory effect on carbon emissions over time. In general, the total effect of each factor inhibited carbon emissions from 2000 to 2005 and promoted carbon emissions from 2005 to 2010, 2010 to 2015, and 2015 to 2020. The contributions of the inhibitory effect and the promoting effect from 2000 to 2020 could roughly offset each other, such that the critical point of each contribution was attained.

#### 4.4.2. Decoupling Efforts of the Driving Factors

The decoupling effort index for each factor was obtained based on the decoupling effort model. As shown in Table 5, the efficiency of technological innovation and the rate of intensive land use made strong efforts for decoupling. The decoupling effort index for the efficiency of technological innovation at first decreased and then increased, reaching a peak from 2015 to 2020, and the efficiency of technological innovation played a crucial role in the ideal decoupling between net carbon emissions and construction land. The decoupling effort index for the rate of intensive land use decreased in volatility, and the efforts weakened. The intensity of carbon emission evolved from strong decoupling efforts to no decoupling efforts in 2015–2020. Similarly, the size of the population also evolved into no decoupling efforts in 2015–2020, indicating that the increase in the intensity of carbon emission and the agglomeration of the population brought about by construction land in recent years weakened the efforts for decoupling. Conversely, the degree of land use evolved from no decoupling efforts to strong decoupling efforts in 2015–2020, confirming that the reduction of construction land can drive the evolution of the decoupling relationship towards the ideal state. The economic scale, industrial structure, and the intensity of technological innovation did not affect the decoupling efforts. Judging from the total decoupling effort index, only the period 2000–2005 exhibited strong decoupling efforts, and the other periods showed no decoupling efforts. This suggested that the various factors did not make sufficient efforts for decoupling, and the policy for carbon reduction in Shandong Province was not properly implemented.

To analyze the spatial distribution of the decoupling efforts for each factor, the effort index for each factor from 2000 to 2020 was studied from a spatial perspective. As shown in Figure 10, there was significant spatial heterogeneity in the degree of the decoupling efforts across the factors from 2000 to 2020. The rate of intensive land use made strong efforts for decoupling throughout the whole province, indicating that the intensive use of construction land in Shandong Province implied that outstanding efforts were made to reduce carbon emissions. It is clearly necessary, however, to improve the intensive use of land continuously. The areas where the size of the population made strong efforts towards decoupling were widely distributed in space, and the population agglomeration effect in most areas contributed to the decoupling. Jinan’s Shizhong District, Tianqiao District, Huaiyin District, Lixia District; Yantai’s Zhifu District, Laishan District; Jimo District in Qingdao; Lanshan District, and Luozhuang District in Linyi; Shizhong District of Zaozhuang had weaker efforts. The six municipal districts of Qingdao, Hekou District, Lingcheng District, Linyi County, Wudi County, and Shouguang City made no efforts. Areas with weak decoupling efforts and no decoupling efforts remind us of the need to reasonably control the scale of population agglomeration so that the population is more evenly distributed in different regions. The strong decoupling efforts linked to efficiencies in technological innovation were concentrated mainly in the municipal districts, with the municipal districts as the core in a “cluster” distribution pattern. The efficiency of technological innovation for most county-level cities and counties did not facilitate decoupling, and these areas should improve their efficiencies in technological innovation by relying on innovation initiatives in municipal districts.

From the perspective of the intensity of carbon emissions, the areas with strong decoupling efforts and weak decoupling efforts were scattered across the province, mostly in municipal districts, and spatially distributed in a “dotted” manner according to the prefecture-level cities to which they belong. These areas made efforts to reduce carbon emissions, but such efforts did not become sufficiently focused, thus, the contradiction between economic development and carbon emissions was still apparent. For these areas, the transformation of the economic development mode should be accelerated. In terms of industrial structure, only Donggang District, Lanshan District, Wulian County, Ju County, and Changdao County made efforts for strong decoupling, while Muping District, Zhaoyuan City, and Rushan City made weak efforts for decoupling, and most of the remaining areas did not make efforts to adjust. This showed that the development of non-agricultural industries in Shandong Province was not conducive to the realization of carbon reduction, therefore, there is an urgency to promote the transformation and upgrading of the industrial structure. From the perspective of land use, only a small number of areas have been striving to achieve decoupling, and Shandong Province needs to regulate the expansion of construction land to reduce the risk of increased carbon emissions. The economic scale and the intensity of technological innovation in the whole province have not contributed to decoupling, and it is necessary to achieve low-carbon and high-quality development to deal with increased carbon emissions caused by the rapid economic development in various regions. The positive role played by the intensity of technological innovation is far from sufficient. In the future, it is essential to increase investment in research and development concerning energy-saving and carbon-reduction technologies.

## 5. Discussion

### 5.1. Accuracy of the Estimation of Net Carbon Emissions

In terms of estimating carbon emissions from energy consumption, the “China Energy Statistical Yearbook” only counts energy consumption data at the provincial level, and it is difficult to obtain energy consumption data at the county level. Although the CEAD has published carbon emissions from energy consumption at the county level in China, the data are only available up to 2017, and the publication of the carbon emissions data is not timely [79]. To ensure the timeliness and uniformity of the research data, we estimated the carbon emissions at the county level from 2000 to 2020 based on the carbon emissions from energy consumption and night-time light data in Shandong Province. Due to the different fitting and correction methods of DMSP/OLS and NPP/VIIRS night-time light images in existing studies, the types of energy chosen for the measurement of carbon emissions from energy consumption also differ [80,81], resulting in differences in the accuracy of estimation of carbon emissions. Even if a double fitting accuracy test was performed, the errors in estimation were still unavoidable. Therefore, to verify the accuracy of the estimation of carbon emissions, the data were compared with the carbon emissions in Shandong Province from 2000 to 2017 released by the CEAD. The verification revealed that the MRE for the two datasets was 15.450%, and the Root Mean Square Error (RMSE) was 98.332 million tons, which shows that the estimation of carbon emissions from energy consumption was accurate. However, Shandong is a coastal province, and the estimation of carbon emissions from land use only factors in the carbon sequestration of land cover and does not consider the carbon sequestration capacity of the ocean. In the future, it will be necessary to further explore the estimation method of the ocean carbon sink, thereby making the estimation of net carbon emissions more accurate and scientifically sound.

### 5.2. Mechanism of Action of Driving Factors on Decoupling

The analysis above demonstrates that for the estimation of net carbon emissions, the total amount of carbon sources far exceeded the number of carbon sinks, and the carbon emissions of energy consumption carried by construction land were the main carbon sources, and this finding is consistent with previous studies [82,83,84]. However, by comparison, our analysis of the decoupling between net carbon emissions and construction land reveals that in the first three periods, the growth rate of net carbon emissions was faster than that of construction land. The period 2015–2020 featured a strong negative decoupling where construction land was shrinking, and net carbon emissions were increasing. Studies have confirmed that the contribution of land use to net carbon emissions was relatively low in each period. This indicated that the expansion or reduction of construction land has a certain impact on net carbon emissions. From 2015 to 2020, the construction land shrank, the net carbon emissions increased, and the degree of land use showed an inhibitory effect on carbon emissions and contributed to decoupling. This means that the reduction of construction land had a positive impact on reducing net carbon emissions, but such an impact was not significant compared with other factors. Based on the results of the analysis of the drivers, this study summarizes in a systematic way the factors influencing net carbon emissions and the mechanism for the role of each factor in decoupling net carbon emissions from construction land. As shown in Figure 11, the driving factors interact with each other and constrain each other in a dynamic equilibrium situation. In the process of urbanization, the reduction of construction land in Shandong Province has started to pay for achieving the ideal decoupling condition. However, the level of the density of the population supported by construction land, the level of economic development, and the carbon emissions arising from economic development have hindered the ideal decoupling condition and are not conducive to sustainable urban development. At the same time, the industrial structure and the level of science and technology innovation in Shandong province are not sufficient to facilitate the desired decoupling of construction land from net carbon emissions; and this is also not conducive to achieving sustainable urban development. On the contrary, the efficiency of science and technology innovation and the intensive use of construction land have constituted efforts to achieve the desired decoupling and have contributed positively to the balance between urbanization and carbon emissions. In addition to controlling the uncontrolled expansion of construction land, local governments in Shandong Province should also make more efforts towards adjusting the economic development model, scientific and technological innovation, transformation of the industrial structure, the size of the population, and intensive land use when formulating targeted and refined reduction measures for carbon emissions.

### 5.3. Contributions of Research Findings

The method for estimation of improved net carbon emissions proposed in this study provides new data to support the long-term dynamic monitoring of carbon emissions at fine scales. A more scientific approach is clearly needed to address the difficulty of estimating carbon emissions at fine scales due to the lack of energy statistics. The basic data required for estimating net carbon emissions are mainly land use remote sensing monitoring data, night-time light data, and energy consumption data. Both land-use remote sensing monitoring data and night-time lighting data are raster data, which can meet the needs of different research scales. Therefore, the method is not only applicable at the county level, but can also be readily extended to the estimation of carbon emissions at different research scales, including the net carbon emissions of other provinces and the national scale.

The results of the study showed that in recent years, net carbon emissions and construction land in Shandong Province have been in a state of strong negative decoupling, and the decoupling relationship was not conducive to achieving reduction targets in carbon emissions. However, the reduction of construction land has started to have a positive impact on reductions of carbon emissions, but not a great extent. The social and economic activities carried out on construction land are the main sources of carbon emissions. Low-carbon green development involves mutual constraints and interactions among various factors. Additionally, studies have shown that urbanization can lead to a large but transient carbon sink, but only a decrease in carbon emissions from fossil fuel burning will make the goal of carbon neutrality achievable [85]. Therefore, in the process of urbanization, it is unwise for local governments to achieve the “carbon peaking and carbon neutrality” goals by blindly reducing the area of construction land if they want to pursue short-lived low carbon development initiatives. The study findings not only contain valuable new information for the local governments of Shandong Province, but also provide useful background and enlightenment for the sustainable development of urbanization and carbon emissions in other countries and regions.

## 6. Conclusions

### 6.1. Conclusions

(1)From 2000 to 2020, the net carbon emissions in Shandong Province continued to increase. The carbon emissions for energy consumption carried on construction land were the main carbon sources, the total carbon sources far exceeded the carbon sinks. Spatially, areas with high carbon emissions tended to from clusters centering on municipal districts, and in the case of Jinan and Qingdao, two distinct carbon emission cluster centers were formed.(2)The first three periods featured an expansive negative decoupling between net carbon emissions and construction land in Shandong Province, and this evolved into a strong negative decoupling from 2015 to 2020. Spatially, the areas with expansive negative decoupling dominated the province. The number of areas with strong and weak decoupling increased from 2005 to 2010, and the number of areas with strong negative decoupling increased from 2015 to 2020. In general, the current decoupling between net carbon emissions and construction land in Shandong Province is not conducive to carbon reduction.(3)From 2000 to 2020, the promoting effect of the economic scale on net carbon emissions was strengthened, while that for the intensity of technological innovation weakened. The inhibitory effect on net carbon emissions due to the efficiency of technological innovation was strengthened, whereas that for the rate of intensive land use weakened. The role of the intensity of carbon emissions and the size of the population evolved from an inhibitory one to a promoting one, and the industrial structure and the degree of land use evolved from a promoting role to an inhibitory role. In general, carbon emissions were promoted, and carbon emissions were inhibited by various factors which can basically offset each other.(4)From 2000 to 2020, the rate of intensive land use and the efficiency of technological innovation made strong efforts with respect to achieving decoupling. Spatially, the rate of intensive land use in various regions of the province strived to achieve the ideal decoupling, and the regions where technological innovation efficiency contributed to decoupling were distributed in clusters centering on the municipal districts. The intensity of carbon emissions evolved from strong decoupling efforts to no decoupling efforts; the areas that strived to achieve decoupling were mainly municipal districts and were distributed in a “dotted” shape. The size of the population evolved from strong decoupling efforts to no decoupling efforts, the areas with strong decoupling efforts were mostly located in county-level cities and counties, while the areas with weak decoupling efforts were mostly distributed in municipal districts. The degree of land use changed from no decoupling efforts to strong decoupling efforts, and, in recent years, the reduction of construction land contributed to ideal decoupling. In general, more efforts are needed, through the involvement of the above factors, to realize an ideal decoupling condition.

### 6.2. Implications

This paper proposes targeted carbon reduction measures to help Shandong Province achieve the goals of “carbon peaking and carbon neutrality” based on the conclusions presented in the last section.

There is a need to improve the efficiency of utilization of land resources and prevent the disorderly expansion of construction land. The study has found that the intensive use of land effectively inhibited carbon emissions, and the inhibitory effect of the degree of land use begins to appear. The Shandong Provincial Control Standards for Intensive Use of Construction Land implemented in 2019 has revised its policy regarding the extensive use of industrial land and rural residential land in construction land, by providing strong policy support for carbon reduction. Under the guidance of these policies, it is crucial to fully realize and utilize the development potential of land resources and limit land supply for high-carbon emission industries, thus favoring larger-scale economic construction with less consumption of land resources. It is also necessary to properly control the scale and speed of the expansion of construction land, thereby avoiding the over-occupying of other land types that serve as carbon sinks.

There is a need to transform the mode of economic development and promote the optimization and upgrading of the industrial structure. This study has confirmed that economic scale plays a significant role in promoting carbon emissions, while, in recent years, industrial structure has somewhat inhibited carbon emissions. Therefore, the effective way to reduce carbon emissions is to transform the economic development mode, reduce excessive dependence on energy consumption, change the structure of energy consumption, and improve the efficiency of energy utilization, thereby reducing the intensity of carbon emissions. It is important to eliminate unproductive capacity that does not meet green and low-carbon development criteria, issue guidelines to high-energy-consuming industries on how to save energy and reduce carbon emissions and promote the optimization and upgrading of industrial structure.

Attention should be paid to the role of scientific and technological innovation the and vigorous development of green and low-carbon technologies. This study has revealed that the intensity of technological innovation has no significant inhibitory effect on carbon emissions, but the efficiency of technological innovation contributes to carbon reduction. In the future, it is essential to increase financial expenditure and invest in energy-saving and emission-reduction technologies. The present study also showed that most of the areas where efficiencies in technological innovation contributed to strong decoupling are mainly municipal districts. Research institutes and universities are concentrated in these regions, and the construction of scientific and technological innovation platforms and the transformation efficiency of scientific and technological achievements are a strong driver for achieving reductions in carbon emissions. By comparison, county-level cities and counties do not contribute to decoupling, indicating that the transfer of scientific and technological innovations to municipal districts is insufficient. It is necessary to promote interactions of scientific and technological innovation in regional development, optimize the innovation environment, and share scientific and technological resources.

### 6.3. Limitations and Proposals for Future Research

In general, there are still some limitations to this study and thoughts and suggestions are given for future research. In terms of estimating net carbon emissions, this study has provided a scientific method for estimating carbon emissions at fine scales. However, the night-time light brightness presents an approximately linear growth pattern, and the growth rate of carbon emissions has increased in recent years. Therefore, reconciling and addressing this practical problem is an important breakthrough direction, whereby we could use night-time light data to simulate carbon emissions in the future. In terms of the decoupling relationship, the Tapio decoupling coefficient was chosen to examine the relationship between net carbon emissions and construction land. However, there are differences in resource endowments, economic levels, and the intensity of energy consumption among counties in Shandong Province. The suitability of the decoupling classification types for the study area also needs to be further tested. In terms of the driving factors, it is difficult to obtain long-term series data at the county level, for example, the capital investment in scientific and technological research and development, and the energy consumption of different production sectors, etc. Additionally, there are limitations to the selection of indicators such as the intensity of technological innovation and the efficiency of technological innovation, and the mechanism underlying the driving factors needs to be better explained.

According to China’s “Code for Classification of Urban Land Use and Planning Standards of Development Land”, construction land can be subdivided specifically into eight types of land use. For example, residential land, industrial land, commercial and business facilities land, etc. This study only analyzed the decoupling relationship between construction land and net carbon emissions, without exploring further the relationships between the specific classifications of construction land and net carbon emissions. In the future, more in-depth research should be conducted with the aid of high-resolution carbon emissions data, statistics for urban construction land, and Point of Interest (POI) data.

## Figures and Tables

**Figure 1 ijerph-19-08910-f001:**
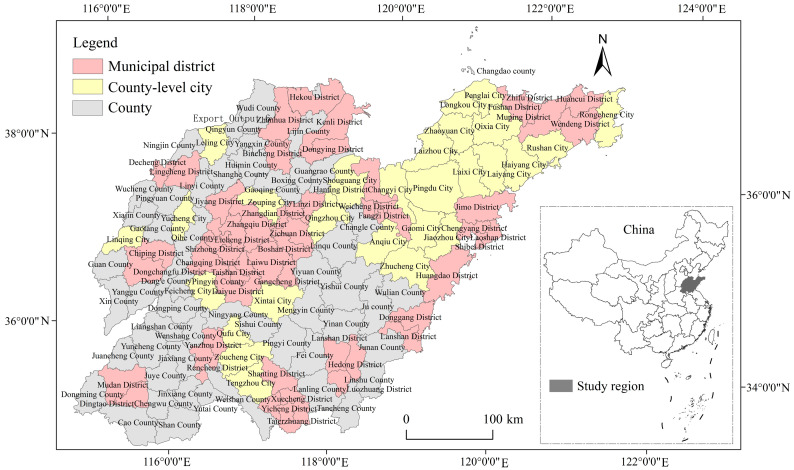
The location of Shandong Province in China.

**Figure 2 ijerph-19-08910-f002:**
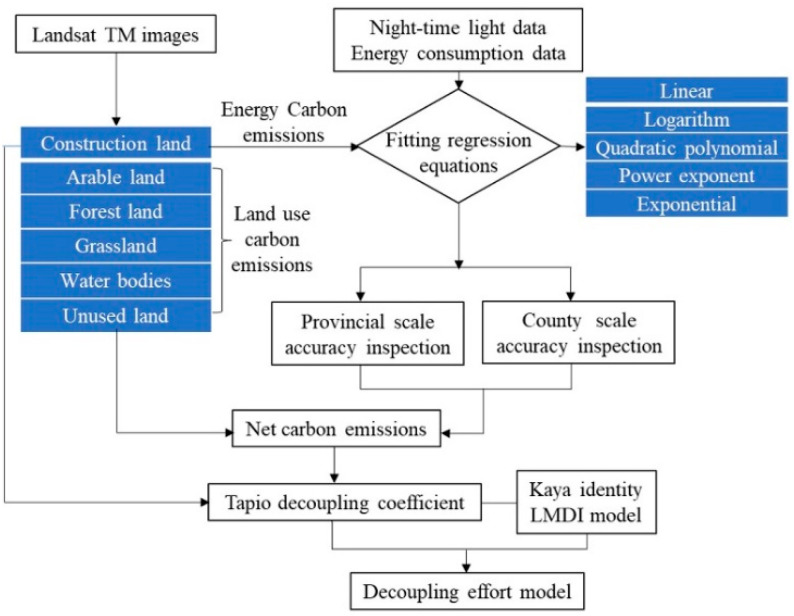
Flowchart of the methodology.

**Figure 3 ijerph-19-08910-f003:**
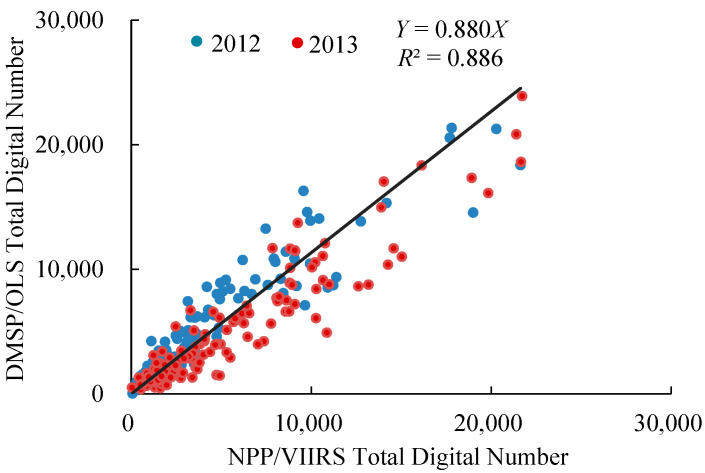
Fitting relationship between the DMSP/OLS and the NPP/VIIRS between 2012 and 2013.

**Figure 4 ijerph-19-08910-f004:**
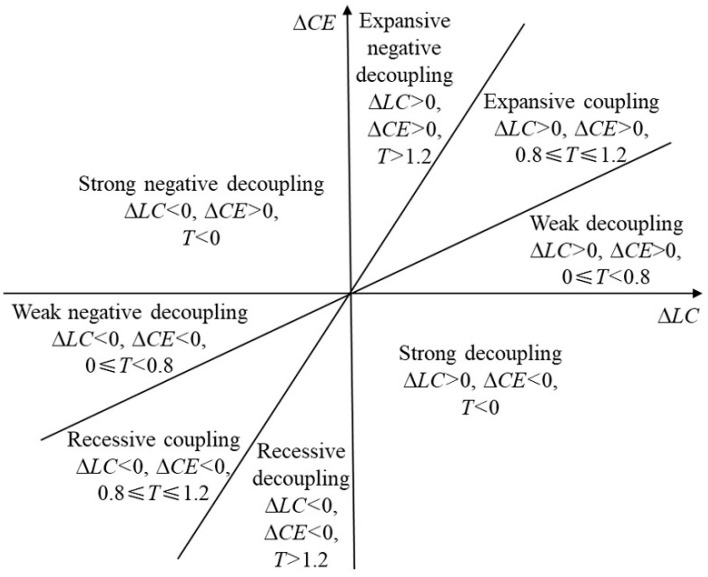
Classification criteria for decoupling relationships.

**Figure 5 ijerph-19-08910-f005:**
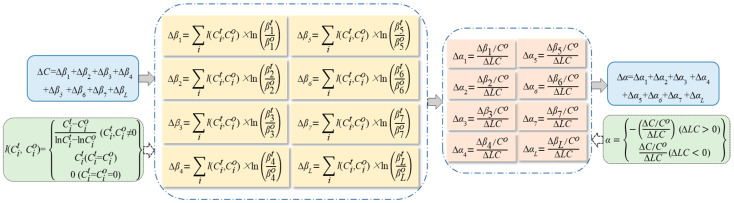
The flowchart for the decoupling effort model.

**Figure 6 ijerph-19-08910-f006:**
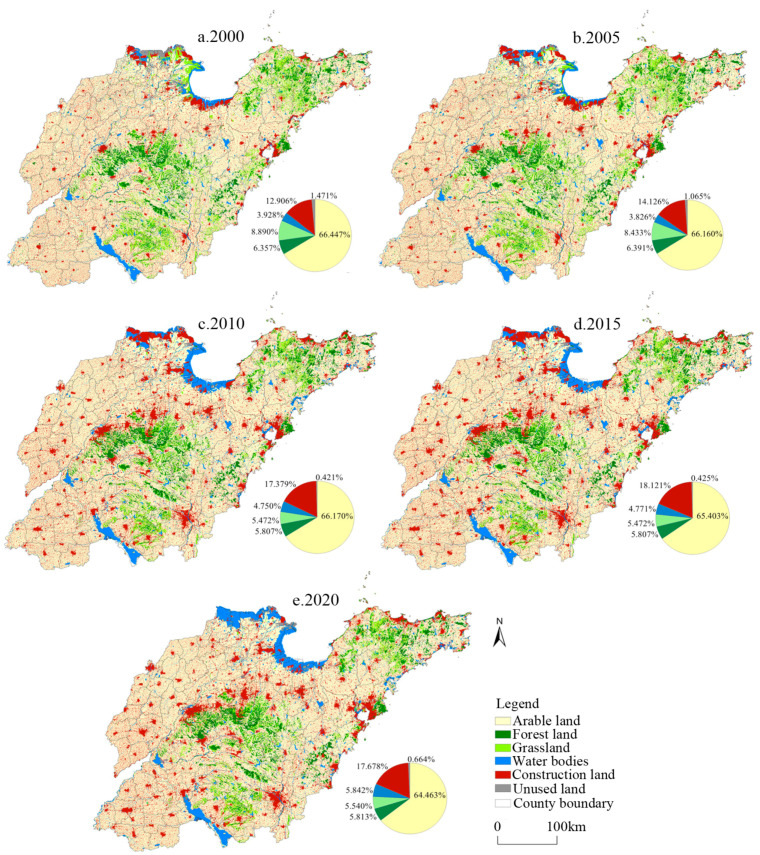
Spatial distribution patterns of land types in Shandong Province.

**Figure 7 ijerph-19-08910-f007:**
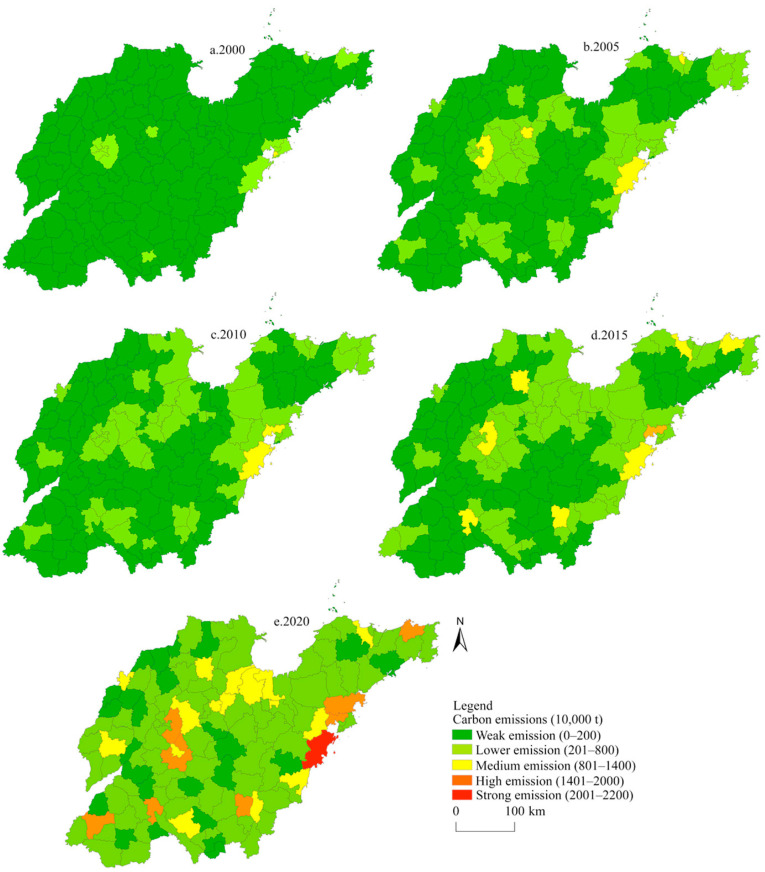
Spatial evolution pattern for net carbon emissions in Shandong Province from 2000 to 2020.

**Figure 8 ijerph-19-08910-f008:**
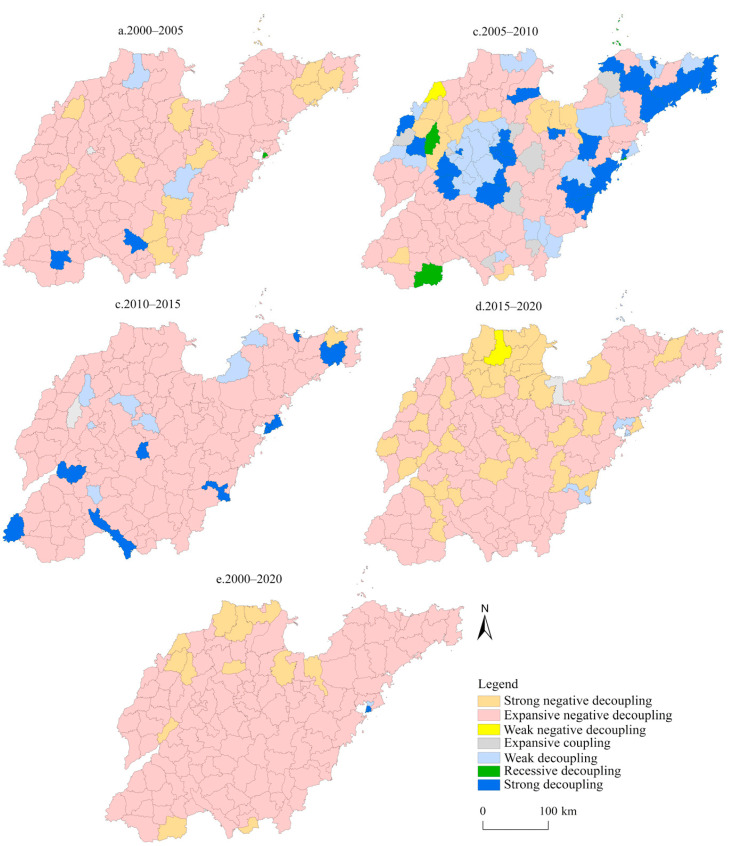
Spatial evolution pattern of decoupling between net carbon emissions.

**Figure 9 ijerph-19-08910-f009:**
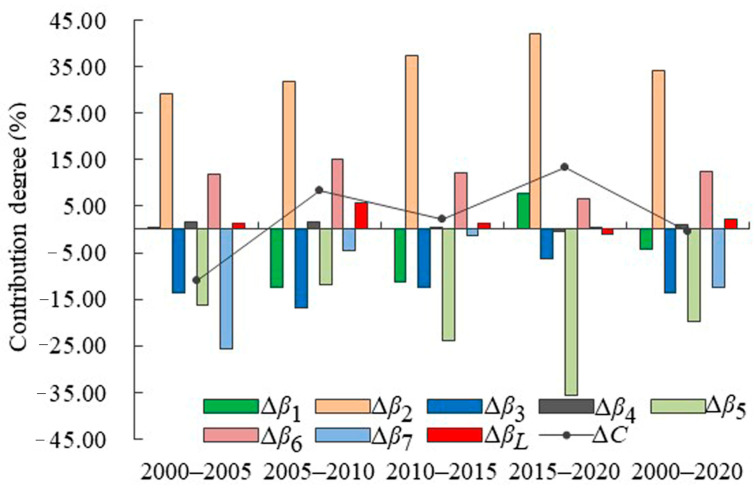
Relative contributions of driving factors.

**Figure 10 ijerph-19-08910-f010:**
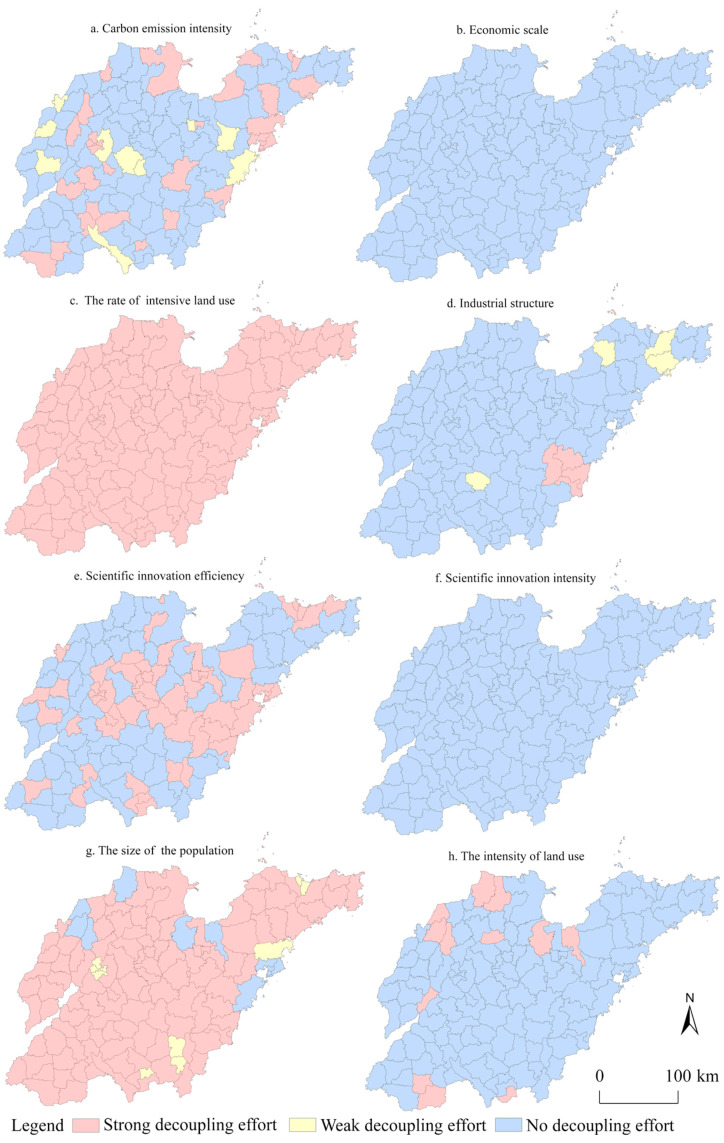
Spatial distribution pattern for decoupling effort index of driving factors.

**Figure 11 ijerph-19-08910-f011:**
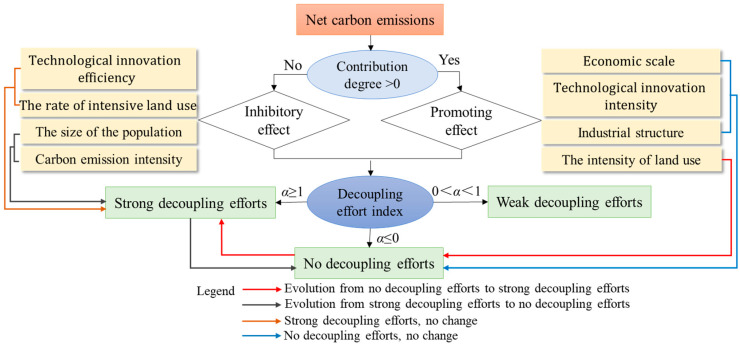
The mechanism of action of the driving factors.

**Table 1 ijerph-19-08910-t001:** Carbon emission coefficients for energy consumption.

Types of Energy	Standard Coal Coefficient	Carbon Emission Coefficient	Types of Energy	Standard Coal Coefficient	Carbon Emission Coefficient
Raw coal	0.714	0.756	Natural gas	1.330	0.448
Coke	0.971	0.855	Heating power	0.034	0.670
Crude oil	1.429	0.586	Electricity	0.345	0.272
Petrol	1.471	0.554	Finished coal	0.900	0.756
Paraffin	1.471	0.571	Coke oven gas	0.614	0.355
Diesel	1.457	0.592	Liquefied petroleum gas	1.714	0.504
Fuel oil	1.429	0.619	Refinery gas	1.571	0.460

**Table 2 ijerph-19-08910-t002:** Fitting equation for carbon emissions from energy consumption.

Model Categories	Fitting Equation	*p*	*R* ^2^	Provincial Scale MRE (%)	County Scale MRE (%)
Linear	Y = 0.058X + 16846.745	0.000	0.764	20.075	26.726
Logarithm	Y = 35034.163lnX − 413160.777	0.000	0.865	14.362	/
Quadratic polynomial	Y = 0.152X − 6.618 × 10^−8^X^2^ − 13347.298	0.000	0.877	12.788	55.265
Power exponent	Y = 0.428X^0.875^	0.000	0.856	15.190	76.165
Exponential	$Y=20611.685e^{X1.287\times 10^{-6}}$	0.000	0.691	23.190	/

**Table 3 ijerph-19-08910-t003:** Carbon emissions of different land types in Shandong Province (10,000 t).

Carbon Emissions	2000	2005	2010	2015	2020
Construction land	17,320.64	42,125.053	57,167.141	61,531.094	79,969.061
Arable land	436.832	432.556	432.635	427.608	422.824
Forest land	−63.781	−63.761	−57.94	−57.944	−58.182
Grassland	−2.908	−2.744	−1.78	−1.78	−1.808
Water bodies	−15.175	−14.701	−18.252	−18.331	−22.518
Unused land	−0.115	−0.082	−0.033	−0.033	−0.052
Total carbon sinks	−81.979	−81.289	−78.005	−78.088	−82.560
Total carbon sources	17,757.472	42,557.609	57,599.776	61,958.702	80,391.885
Net carbon emissions	17,675.493	42,476.321	57,521.771	61,880.614	80,309.325

**Table 4 ijerph-19-08910-t004:** Decoupling between net carbon emissions and construction land.

Study Period	Δ*LC*	Δ*CE*	*T*	Decoupling Relationships
2000–2005	0.088	1.403	15.856	Expansive negative decoupling
2005–2010	0.230	0.354	1.538	Expansive negative decoupling
2010–2015	0.043	0.076	1.775	Expansive negative decoupling
2015–2020	−0.021	0.298	−13.986	Strong negative decoupling
2000–2020	0.367	3.544	9.664	Expansive negative decoupling

**Table 5 ijerph-19-08910-t005:** Decoupling effort index of driving factors.

Study Period	$\Delta \alpha_{1}\;$	$\Delta \alpha_{2}$	$\Delta \alpha_{3}$	$\Delta \alpha_{4}$	$\Delta \alpha_{5}$	$\Delta \alpha_{6}$	$\Delta \alpha_{7}$	$\Delta \alpha_{L}$	Δα
2000–2005	1.377	−22.174	10.315	−1.209	12.292	−9.105	19.635	−1.330	8.046
2005–2010	2.153	−5.519	2.911	−0.285	2.083	−2.626	0.811	−2.026	−2.499
2010–2015	8.830	−29.486	9.927	−0.338	18.843	−9.589	1.055	−2.157	−2.917
2015–2020	−8.052	−39.206	4.743	−0.618	31.331	−7.125	−1.249	2.533	−17.644
2000–2020	2.022	−16.261	6.456	−0.536	9.469	−5.919	5.992	−1.735	−0.514

## Data Availability

The data presented in this study are available on request from the corresponding author. The data are not publicly available because research is ongoing.

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
