# Peer review of "Estimating the Decoupling between Net Carbon Emissions and Construction Land and Its Driving Factors: Evidence from Shandong Province, China"

_ijerph, 2022, doi:10.3390/ijerph19158910_

Round 1

Reviewer 1 Report

There is a need to make clear the research objective in the introduction. Perhaps defining a research question will improve this understanding.
It is necessary to better develop the Literature Review section, present what other similar studies have done in the area, end the section with a critical analysis paragraph and, if possible, define a research hypothesis.
It is necessary to better justify the choice of method of treatment of the data obtained, why is it compared to other existing ones that guarantee the same results? How was the area analyzed in the study defined? How important is it?
In discussions, clearly and objectively include contributions of study findings to theory and practice.
Highlight research limitations and proposals for future studies in the conclusions.

Reviewer 2 Report

This paper has systematically summarized the influencing factors of net carbon emissions and the mechanism of each factor on the decoupling between net carbon emissions and construction land. I think this paper tackles a highly important and interesting research topics and uses the resourceful quantitative methodologies to answer the research question, but there are some places for the research to improve with minor revision before publication.

In the first place, the paper intends to make the three contributions. First, we factor in the carbon source and carbon sink capacities of different land types and to conduct empirical research using net carbon emissions. Second, the research scale has expanded to the finer county scale. Finally, the Tapio decoupling coefficient, Kaya identity, and LMDI model was integrated to analyze the influencing factors of net carbon emissions and the driving effect of each factor on the decoupling between the two. Overall, the contributions are significant and the methodology is easy to understand. However, there are minor improvement suggestions to consider. The limitations of the methodology should be addressed and acknowledged. There is limitations on the decoupling coefficient concerning the relationship between the carbon emissions. The arguments concerning the technological innovation intensity and technological innovation efficiency is hardly and partly insufficient to support the argument. The results are interpreted superficially and not doing justice to the set-relational and configurational nature of such results.

For these reasons, I suggest the minor revision of the paper before the publication.

Reviewer 3 Report

The paper analyses the potential relation between net carbon emissions and construction land. The theme is relevant, the research is highly detailed, the materials and methods are justified and seem adequate and the relation with previous literature is done.

My suggestions are:

1.       Add a brief description of the dataset and the main findings of the paper at the end of the Introduction. The information on the unique contribution of the paper, presented in the last paragraph of the introduction, should be therefore enriched.

2.       The document should have a proper discussion of the extent to which the results may be generalized to other environments and settings than Shandong Province. Such discussion is fundamental to future scientific research.

3.       The limitations of the study and the suggestions for future research should be included in the Conclusion section.

4.       Variables Y and X in line 245 need to be properly defined. Also, add an interpretation of the respective regression results.

Some additional remarks go to text adjustments and proofread:

1.       The ideas are clear, but the text would benefit if it were more concise. Avoid using long paragraphs as they hamper reading (e.g., p. 13 and 16).

2.       Some acronyms are used but have not been defined (e.g., DMSP/OLS; NPP/VIIRS).

3.       Typos. E.g., in line 71 replace “very” with “vary”.

Reviewer 4 Report

A minor language editing is required

Round 2

Reviewer 1 Report

The paper is ready for publication in its current form.

Reviewer 3 Report

The main concerns previously raised have been adequately addressed by the authors. Congratulations to the authors for their revised version.